# Bacteriophage Delivery Systems Based on Composite PolyHIPE/Nanocellulose Hydrogel Particles

**DOI:** 10.3390/polym13162648

**Published:** 2021-08-09

**Authors:** Tilen Kopač, Ana Lisac, Rok Mravljak, Aleš Ručigaj, Matjaž Krajnc, Aleš Podgornik

**Affiliations:** 1Department of Chemical Engineering and Technical Safety, Faculty for Chemistry and Chemical Technology, University of Ljubljana, SI-1000 Ljubljana, Slovenia; tilen.kopac@fkkt.uni-lj.si (T.K.); ana.lisac@fkkt.uni-lj.si (A.L.); rok.mravljak@fkkt.uni-lj.si (R.M.); Ales.Rucigaj@fkkt.uni-lj.si (A.R.); matjaz.krajnc@fkkt.uni-lj.si (M.K.); 2COBIK, Tovarniška 26, SI-5270 Ajdovščina, Slovenia

**Keywords:** TEMPO nanocellulose, T7 bacteriophage, encapsulation, drug diffusion, zero-order release

## Abstract

The role of bacteriophage therapy in medicine has recently regained an important place. Oral phage delivery for gastrointestinal treatment, transport through the stomach, and fast release in the duodenum is one of such applications. In this work, an efficient polyHIPE/hydrogel system for targeted delivery of bacteriophages with rapid release at the target site is presented. T7 bacteriophages were encapsulated in low crosslinked anionic nanocellulose-based hydrogels, which successfully protected phages at pH < 3.9 (stomach) and completely lost the hydrogel network at a pH above 3.9 (duodenum), allowing their release. Hydrogels with entrapped phages were crosslinked within highly porous spherical polyHIPE particles with an average diameter of 24 μm. PolyHIPE scaffold protects the hydrogels from mechanical stimuli during transport, preventing the collapse of the hydrogel structure and the unwanted phage release. On the other hand, small particle size, due to the large surface-to-volume ratio, enables rapid release at the target site. As a consequence, a fast zero-order release was achieved, providing improved patient compliance and reduced frequency of drug administration. The proposed system therefore exhibits significant potential for a targeted drug delivery in medicine and pharmacy.

## 1. Introduction

In recent years, the spread of antibiotic-resistant bacteria became a global health problem making potential alternatives like bacteriophages an important field of study [1]. Bacteriophages are natural killers of bacteria, which lyse bacteria to produce new progeny viruses [2]. Phage therapy is intriguing since phage self-replication ability leads to a local increase in their concentration, and the narrow host specificity of phages ensures the lack of broad off-target effects [3,4,5,6]. The success of phage therapy depends on bacterial physiological state and phage stability, survival, and consistent phage titer (concentration) for a stable dosage delivered at the site of infection [7,8]. Oral administration of phages is the least invasive and an easy to handle technique, however, there are some drawbacks since there is a possibility of phage inactivation when passing subsequent gut sections due to the acidic and proteolytic environment of the stomach [9]. Several studies have demonstrated that low pH can affect and inactivate phage populations [10,11]. Phage T7 used in our experiments remains stable at pH 6–8 and is completely inactivated at pH below 3 [12]. To improve phage stability, encapsulation can be employed to increase the retention time of phages, to contain phage concentration at a therapeutically effective level, and to control the time of release [7] but also its stability. For gastrointestinal infections, phages were mostly encapsulated in liposomes [13,14], and for other infections such as burn wounds, biofilms on implants, and root canal infections, phages were encapsulated in microparticles (d = 8.0 ± 4.5 µm) [15], scaffolds [16], and hydrogels [17,18,19], in materials such as biopolymers [20], synthetic and semisynthetic polymers, and inorganic materials [16,21].

One of the rather novel biopolymers with several important properties is nanocellulose [22]. Especially, biopolymer-based hydrogels synthesized with 2,2,6,6-tetramethylpiperidine-1-oxyl (TEMPO)-oxidized cellulose nanofiber (TOCNF) have a great application prospect in medicine and pharmacy as well, especially in drug delivery systems [23,24,25,26]. TOCNF is a relatively new material in this field and could play a major role in future research due to its excellent properties, such as mechanical and chemical stability, non-toxicity, renewability, biodegradability, biocompatibility, and easy accessibility [27]. The best-known hydrogel synthesis is performed by ionic crosslinking with divalent ions, such as calcium(II) ions, which keeps the system non-toxic and suitable for medical and pharmaceutical use [27,28]. The mechanism of hydrogel formation is based on ionic interactions of the carboxyl functional groups present in the TOCNF with pK_a_ 3.9 [29,30]. Their protonation causes hydrogel and pore shrinkage at pH < pK_a_ due to polymer chain attraction, while hydrogel and pore swelling is due to deprotonation at pH > pK_a_ due to polymer chain repulsion [30]. Moreover, unlike sodium alginate, the most commonly used biopolymer in drug delivery [31], TOCNF can be crosslinked without the addition of a crosslinking agent by hydrogen bond formation. Ageing of aqueous TOCNF dispersions at a concentration of 1–3 wt.% can result in hydrogel structure over time. The ageing of TOCNF aqueous dispersions has already been studied in detail [32]. In this case, the biopolymer chains are crosslinked via hydrogen bonds. Secondly, as a result of carboxyl pendant groups on the surface, the TOCNF (anionic biopolymer) has a great possibility to be used for targeted drug delivery [30].

When hydrogels are used as a phage encapsulation system, their particle size is significant. While larger particles provide better protection from the acidic environment [33], smaller particles have shorter retention times through the gastrointestinal tract and faster release [34]. Therefore, it is important to tailor particle size according to the specific application. One possible approach is to introduce hydrogel in rigid, pre-synthesized, porous particles that can be highly diverse in their morphology, chemistry, and stability [35]. Among them, high internal phase emulsion polymer (polyHIPE) particles exhibiting high porosity, mechanical stability, diversity in size, and microstructure consisting of large void pores (VP) and smaller interconnecting pores (IP) connecting VP in all three dimensions [36]. As both the VP interconnectivity and porosity are high, this results in a highly permeable material capable of a high encapsulation capacity and minimal mass transport resistance. While polyHIPE morphology itself already has a good disposition due to the cage-like structure of the larger VP surrounded by smaller IP which can be independently tailored to match particular application [37], and excellent mechanical properties [38] assure robust and reliable performance. We therefore hypothesize that polyHIPE materials are suitable for protection and fast release of the encapsulated drug. Based on the chemistry of the chosen monomers, they can be easily functionalized [39]. High structural variability is achieved by a variety of preparation procedures such as continuous microfluidic photopolymerization [40,41] or via triple emulsion induced by dispersing HIPE inside a solution of a stabilizer causing polydisperse particle formation, which is then polymerized [42,43]. As such, they were already implemented for bone filling and cell delivery [44,45].

In this work, we prepared a polyHIPE–hydrogel composite material with encapsulated T7 bacteriophage for potential application in phage therapy. Immobilization efficiency and release were investigated in an acidic environment with pH such as to mimic conditions that are encountered in the stomach (pH = 2) and duodenum (pH = 5–7) [46] where release is preferred. Release of bacteriophage was studied under different preparation and pH conditions, demonstrating zero-order kinetics.

## 2. Materials and Methods

The experimental work includes the preparation of an efficient system for encapsulation of T7 bacteriophages in a release medium with a pH of 2 and activation of the release at the highest possible release rate in the medium at a pH of 5–7. Two crosslinking mechanisms were used to prepare 1.5% TOCNF hydrogels. First, ageing hydrogen bonding between polymer chains forming a hydrogel 3D structure (crosslinking without the addition of ionic or chemical crosslinkers) and second, ionic crosslinking with Ca^2+^ ions’ addition of 15 mM aqueous CaCl_2_ solution.

### 2.1. Materials

Nanocellulose hydrogel was prepared using 2,2,6,6-tetramethylpiperidine-1-oxyl (TEMPO)-oxidized cellulose nanofibers (TOCNF) with the chemical formula [(C_6_H_10_O_5_)_x_(C_6_H_9_O_4_COONa)_y_] and the carboxylate level of 0.2–2 mmol/g solids that was purchased from The Process Development Centre, University of Maine (UMAINE PDC), USA. Calcium chloride was used as received from Merck (Darmstadt, Germany).

Sodium phosphate dibasic heptahydrate (Na_2_HPO_4_·7H_2_O), sodium dihydrogen phosphate monohydrate (NaH_2_PO_4_ · H_2_O), sodium acetate (CH_3_COONa), sodium hydroxide (NaOH pellets, an assay of ≥98% (acidimetric)) and hydrochloric acid (HCl, an assay of 36.5–38.0%) were supplied by Sigma-Aldrich (St. Louis, MO, USA). Listed chemicals were used as received for preparation of phosphate (pH = 7) and acetate (pH = 5) buffer.

PolyHIPE particles were synthesized using poly(ethylene glycol)-*block*-poly(propylene glycol)-*block*-poly(ethylene glycol) or Synperonic^®^ L 121 (PL121) and Synperonic^®^ F 127 (PF127) (Sigma-Aldrich, St. Louis, MO, USA) as surfactants, glycidyl methacrylate (Sigma-Aldrich) (GMA), ethylene glycol dimethacrylate (Merck KGaA, Darmstadt, Germany) (EGDMA) as monomers, phenyl bis(2,4,6-trimethyl benzoyl)phosphine oxide (BASF Schweiz AG, Kaisten, Switzerland) (IC819) as photoinitiator, calcium chloride dihydrate (Honeywell Fluka, Charlotte, NC, USA) (CaCl_2_·2H_2_O), and ethanol (Kefo, Ljubljana, Slovenia) (EtOH). Deionized water was filtered through a 0.22 µm filter before use.

Bacteriophages were produced with *E. coli* growing, and enumerated using Lysogeny Broth (LB) (LLG Labware, Meckenheim, Germany), agar (Sigma-Aldrich, St. Louis, MO, USA), and SM buffer (1 g gelatin (Sigma-Aldrich, St. Louis, MO, USA), 5.8 g NaCl (Merck KGaA, Darmstadt, Germany), 2 g MgSO_4_∙7H_2_O (Merck KGaA, Darmstadt, Germany), 50 mL 1 M Tris-HCl (pH 7.5) (Fisher Scientific F, Loughborough, UK), and deionized water to 1 L).

### 2.2. Bacteriophage and Bacterial Strains

A lytic *Podoviridae* bacteriophage T7 (DSM 4623) and its host *Escherichia coli* K-12 MG1655 strain (DSM 18039) were used in all experiments (DSMZ Institute, Leibniz, Germany). High titer phage lysate was produced using well-established protocols recommended by DSMZ Institute, Germany. Bacterial cultures for phage titer determination were prepared in laboratory flasks in Lysogeny Broth (LB) (pH 7) and incubated at 37 °C overnight [47].

### 2.3. PolyHIPE Particle Preparation

PolyHIPE particles were synthesized using a similar composition reported previously for polyHIPE monolith preparation [48]. To prepare 30 mL of HIPE, first, the organic phase was prepared in a tall 100 mL beaker consisting of 2.188 g GMA, 1.003 g EGDMA, 0.348 g PL121, and 0.016 g IC819. The aqueous phase was prepared separately with 26.231 g demi water, 0.262 g CaCl_2_∙2H_2_O, and 0.362 g PF127. The organic phase was then mixed at 8000 rpm (Ultra Turrax mixer, IKA T25, S25N-18G disperser, IKA, Germany), and the aqueous phase was slowly added at a flow rate of 2 mL/min. Initially, formed emulsion exhibited high viscosity, decreasing substantially with further addition of the aqueous phase resulting in fluid water-in-oil-in-water (W/O/W) triple emulsion. The beaker containing emulsion was irradiated between two UV lights (light intensity ≈ 13 mW/cm^2^, wavelength maximum at 365 nm) for 1 h, and then diluted with 20 vol.% EtOH. Before, application particles were filtered through various mesh sizes to obtain particles ranging from 15–50 μm and washed extensively with 20 vol.% EtOH.

### 2.4. PolyHIPE Particle Characterization

Field-emission scanning electron microscope (FEG-SEM, JEOL JSM 7600 F, Jeol Inc., Tokyo, Japan) was used to observe the morphology and size of polyHIPE particles after synthesis. Furthermore, images were analyzed with ImageJ software to determine the polyHIPE particles interconnecting pore size and distribution. PolyHIPE particle size analysis after filtration was performed with an optical microscope (Zeiss Imager.Z2m, Carl Zeiss AG, Jena, Germany) in reflected light.

### 2.5. Preparation of PolyHIPE/Hydrogel Particles with Encapsulated T7 Bacteriophage

Particles containing hydrogel and bacteriophage were prepared in 10 mL of suspension containing polyHIPE particles and T7 bacteriophage at a concentration of 2 mg/mL and 1.85 × 10^9^ PFU/mL, respectively, in 10 mM phosphate buffer (pH = 7). The suspension was mixed overnight on a shaker to allow T7 bacteriophage diffusion into the polyHIPE particles. During the mixing, the CaCl_2_ acting as an ionic crosslinking agent was added to the dispersion at a final concentration of 15 mM. Afterward, dry TOCNF was added under constant stirring at 100 rpm to achieve a dispersion concentration of 1.5% (*w*/*v*). The dispersion was then additionally stirred for 24 h.

Particles without CaCl_2_ addition were also prepared using the same procedure, except that at the end, the dispersion was stirred for 7 days to allow hydrogel ageing and hydrogen bond formation [32].

The dispersion was filtered through filter paper (Macherey-Nagel 640 with pore size of 12–25 µm). The filter cake with polyHIPE/hydrogel/bacteriophage particles was washed with 10 mM HCl solution to remove unreacted TOCNF from the particle surface. For better understanding of the paper, the polyHIPE/TOCNF particles’ synthesis are illustrated in Scheme 1.

### 2.6. Phage Titer Determination

Bacteriophage concentration (plaque-forming units; PFU mL^−1^) was determined by standard double agar overlay plaque assay [49]; 50 μL of phage sample was transferred to Eppendorf tube containing 450 μL SM buffer. The dilution series was made from each sample and 10 mL of each dilution in triplicates was dropped on double-layer LB agar plastic Petri dishes with 90 mm diameter; 5 mL of LB with 0.7% agar (*w*/*v*) were mixed with 100 μL of overnight bacterial culture and then poured on LB agar plate with 1.4% agar (*w*/*v*) forming bacterial lawn. LB agar plates were incubated at 37 °C overnight and plaques were enumerated after 6 h.

### 2.7. Behavior of Phage Release from PolyHIPE/Hydrogel and the Determination of Kinetics Parameters

A description of solute release behavior of the controlled release hydrogel was determined by Ritger and Peppas equation which relates the fractional drug release (*M*_t_/*M*_0_) to the release time (*t*) [50,51]:(1)$$\frac{M_{t}}{M_{\infty }}=kt^{n}$$
where *M*_t_ is mass of drug released until time *t*, *M*_∞_ is the amount of drug released when equilibrium is achieved, *k* is a constant incorporating characteristics of the macromolecular network system and the drug, and *n* is the diffusional exponent, which is indicative of the transport mechanism. The behavior of the controlled release from hydrogel (diffusional exponent *n*) and the drug release rate (*k*) were obtained from the linear logarithm form of Equation (1):(2)$$\ln(\frac{M_{t}}{M_{\infty }})=\ln k+n\ln t$$
where the values of *k* and *n* were obtained from the intercepts and gradients of the plots of ln(*M*_t_/*M*_∞_) versus ln(*t*).

## 3. Results and Discussion

### 3.1. Synthesized PolyHIPE Particles

To prepare the desired triple W/O/W emulsion HIPE, two nonionic surfactants namely PL121 and PF127 with hydrophilic-lipophilic balance (HLB) values of 1 and 22, respectively, were used [52]. Their ratio was found to be very important to obtain stable particles of required dimensions since compositions departing from the optimal ratio produced no particles. To accommodate as much as possible hydrogel and bacteriophages, mechanically stable polyHIPE particles with 90% porosity were prepared and further characterized. The density of the polymer particles was calculated to be 0.11 g/mL using the porosity and bulk polymer density (1.1 g/mL). Figure 1 shows particles exhibiting size distribution predominantly from 10 to 50 μm with an average diameter of 24 μm and FWHM = 21 μm.

### 3.2. Targeted Release of Bacteriophage T7 from 1.5% TOCNF Hydrogel

PolyHIPE/hydrogel/bacteriophage composite was prepared in two different ways. The first approach is rather intuitive using Ca^2+^ ions as a crosslinking agent causing ionic interactions and rapid hydrogel formation. The second one was based on differences in the shear force fields. While formation of the hydrogel 3D structure in the dispersion and on the surface of polyHIPE particles is restricted due to a constant stirring, this is not the case inside the polyHIPE particles where the liquid is stagnant, consequently allowing the formation of a hydrogel structure. The polyHIPE particles were therefore not only beneficial to define delivery system geometry but also enabled the preparation of composite without the addition of any crosslinking agent. In this case, the effect of ageing of TOCNF to form hydrogel structure was exploited [32]. To better understand the mechanical properties of TOCNF hydrogels, which play a pivotal role also in the case of bacteriophage delivery, rheological properties were studied in detail in our recent articles [28,30]. The average pore size of TOCNF hydrogels was determined based on oscillatory frequency sweep tests. The huge amount of hydrogel samples (different concentration of biopolymer and crosslinker) tested in [30] allowed to choose the most suitable TOCNF hydrogel composition for phage encapsulation and pH trigger release.

The release from both composites was investigated at 37 °C by changing the pH of the release environment from 2 to 5 or 7. Such conditions were chosen to simulate conditions present in the human body. Results in Figure 2 (inset) show that bacteriophages are not released from the hydrogel at pH 2, irrespective of the crosslinking mechanism used to prepare the hydrogel. T7 bacteriophages have a hydrodynamic diameter of approximately 60 nm [53]. As shown in our previous work [30], the average pore size of 1.5% TOCNF hydrogel in a medium with pH < pK_a_ is 11.4 nm (0 mM Ca^2+^) and 7.9 nm (15 mM Ca^2+^). The smaller pore size in the hydrogel compared to the hydrodynamic radius of T7 encapsulates bacteriophages in the hydrogel without allowing diffusion. After 3 h, which is the average time needed to empty the stomach by 50% [54], the medium was changed to higher pH values (pH of 5 or 7). Figure 2 shows the release of bacteriophages when the pH of the medium was changed which was in less than 30 min. The release activated by the pH change above the substituent group pK_a_ of TOCNF is enabled due to electrostatic repulsive forces between the negatively-charged polymer chains as a result of deprotonation, which caused movement of polymer chains and enlargement of the pores in the hydrogel network [30,55].

A more detailed analysis of bacteriophage release from differently crosslinked samples is shown in Figure 3. The linear dependence of the released bacteriophage amount normalized to the total mass estimated from the mass balance as a function of time is demonstrated indicating a linear release. From the comparison of black and green lines, it can be concluded that release is independent of the pH value once it is higher than the pK_a_ of the substituent groups. On the other hand, it can be observed that the release rate is a function of the crosslinking mechanism.

The release is faster in a hydrogel without a crosslinking agent (Figure 3) [30]. Due to stronger repulsive electrostatic interactions in comparison to predominant hydrogen bond interactions between polymer chains [30,32,56], the hydrogel prepared under ageing swells faster, resulting in larger pores in a shorter time interval than in the case of an ionically crosslinked hydrogel. In this case, ionic interactions are mainly involved in the crosslinking process along with hydrogen interactions [56]. A higher crosslinking density is achieved, which consequently induces smaller pores in the hydrogel [28]. Therefore, during deprotonation, the repulsive electrostatic interactions make it difficult to move polymer chains that are interconnected by stronger ionic interactions. On the other hand, a very low concentration of the ionic crosslinking agent (15 mM) was used, so the differences in the bacteriophage release rate from the different crosslinked hydrogels are minimal (see Figure 2 and slope of the linear function in Figure 3).

### 3.3. Kinetics Behavior, Mechanisms, and Order of Bacteriophage Release

The release behavior of solutes from TOCNF system incorporated into polyHIPE was investigated by defining the release mechanism considering the assumption that bacteriophages do not interact with the hydrogel. As explained in Section 3.2., phage release did not occur at pH 2 (Figure 2, inset) but it is triggered by pH increase. The release is present when the hydrogel pore size is sufficiently increased due to swelling allowing phage diffusion. Accordingly, one would also expect that due to a high polyHIPE porosity, the release is only a function of hydrogel mesh size and as such is controlled by a Fickian diffusion with a high velocity of phage from the matrix and low velocity of polymeric relaxation, similar to what was already shown for TOCNF hydrogels [28,30]. In contrast, the results presented in Figure 3 indicate zero-order release kinetics. This makes us conclude that the release mechanism cannot be described by a simple diffusion model and that a more complex description is required.

One possibility is using the semi-empirical Ritger and Peppas equation (Equation (1)) allowing discrimination between different transport mechanisms. The transport mechanism is determined with the diffusion exponent *n* value, namely for spherical hydrogel particles: (i) *n* = 0.43 (Fickian diffusion, where the diffusion is much greater than the process of polymer chain relaxation), (ii) 0.43 < *n* < 0.85 (anomalous (non-Fickian) transport where the diffusion and relaxation rates of the polymer network are comparable), and (iii) *n* = 0.85 (zero-order release, Case II where diffusion is very fast compared to the relaxation of the polymer network) [50,51]. An extreme example of (iii) is Super Case II (*n* > 0.85) which suggests that the velocity of solvent diffusion is much higher than the polymeric relaxation process, causing an acceleration of solvent penetration [51,57].

The results from Figure 3 were recalculated using Equation (2) and shown in Figure 4. Linear fit allowed determination of *k* and *n* shown in Table 1. It has to be mentioned that R^2^ is satisfactory due to high relative standard deviation of counting method that limits accurate determination of bacteriophage concentration [58,59]. According to literature [50], the Equation (1) is valid for the first 60% of the fractional release. Furthermore, in Figure 4, the linear fit for 100% of the fractional release is evident. In all cases, *n* value is above 1, therefore the phage release can be classified as transport Super Case II, where the phage release mechanism is described as a consequence of swelling and macromolecular relaxation of the polymer chains limiting the typical diffusion process. Apparently, the porous polyHIPE matrix behaves as an obstacle to hydrogel relaxation and swelling. One can imagine that OH^-^ ions penetrating from more basic release medium can freely enter the polyHIPE particle. Next, the repulsive forces cause the movement of the polymer chains, which allows water to diffuse into the hydrogel matrix [30]. In the case of a linear growth of the hydrogel system, the volume of the hydrogel increases to infinity, until the weaker hydrogen and ionic interactions can no longer maintain the 3D structure of the hydrogel [60], forming branched or forked chains with dimensions ranging from 20–50 nm in width and lengths up to several micrometers [32]. They start to concurrently diffuse with the phages into solution through polyHIPE pores of 2.8 µm, additionally affecting the rate of drug release.

The efficiency of the TOCNF-based hydrogel systems lies in the fast release rate followed by the 100% release of bacteriophages. Firstly, even though that phage diffusion is limited, the release time is highly favorable enabling complete release in 30 min due to the small dimensions and a very high surface area to hydrogel volume ratio of the polyHIPE particles. From the lower *k* values for hydrogels prepared by ionic crosslinking, it can be concluded that they swell slightly slower with slower macromolecular relaxation compared to aged hydrogels. As follows, the release rate for systems without ionic crosslinking is slightly faster than in the ionically crosslinked hydrogels which can be concluded from the higher fitted value of *k* (Table 1). Secondly, the complete bacteriophage release proves that the crosslinking of the hydrogel and the encapsulation of the bacteriophages did not affect the stability of phages in hydrogels, which is a potential problem with many other polymer-based hydrogels, but even protected them against low pH value deactivation [61]. Finally, complete dissolution of the hydrogel structure occurs, allowing 100% release of bacteriophages into the release environment, which can be seen after 30 min (Figure 2).

The main purpose of this work was to show the benefits of combining microscopic porous particles with hydrogels for the encapsulation of T7 bacteriophages. We showed that protection against low pH can be achieved and that the targeted release rate is constant until all phages are released which takes only 30 min. Furthermore, over 80% of phages were released in the first 5 min independently of the hydrogel preparation method. Taking also into consideration that ionically crosslinked TOCNF hydrogels have better mechanical and rheological properties, can be easily morphologically altered [28,30,32], and can be prepared faster compared to aged hydrogels, we think that they are more suitable for medical and pharmaceutical applications. Finally, a zero-order release was achieved which is, according to the applications of such delivery systems in pharmacy, the best way to control plasma concentration offering several advantages, including improved patient compliance and reduction in the frequency of drug administration [62].

## 4. Conclusions

In this work, new systems for targeted drug delivery were presented. PolyHIPE/TOCNF hydrogel systems for encapsulation of T7 bacteriophages and targeted delivery to the medium at pH 5–7 was prepared. Protection (encapsulation) of bacteriophages in TOCNF hydrogels allows transport through the stomach to the duodenum. TOCNF hydrogels proved to be an excellent encapsulation system due to their exceptional mechanical properties, as the bacteriophage infectivity did not decrease during crosslinking. Furthermore, polyHIPE particles protected TOCNF hydrogels in their scaffold and prevented its degradation. On the other hand, rapid release activated by pH change followed zero-order kinetics, which is the preferred dynamics of all controlled release drug delivery mechanisms. By preparing very small hydrogel systems in polyHIPE particles resulting in high surface area to hydrogel volume ratio, complete release was achieved after only 30 min. PolyHIPE/hydrogel systems are therefore potentially very useful systems in the field of targeted drug delivery, as hydrogel and polyHIPE morphology and/or chemistry can tailored to specific application. Last but not least, the non-toxicity of such systems also allows their use in the medical and pharmaceutical industries.

## Data Availability

Not applicable.

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
