# Peer review of "Bacteriophage Delivery Systems Based on Composite PolyHIPE/Nanocellulose Hydrogel Particles"

_polymers, 2021, doi:10.3390/polym13162648_

Round 1
Reviewer 1 Report
Comments to the Authors:
This is an interesting and overall well-written paper, this manuscript presents an efficient polyHIPE/hydrogel system for targeted delivery of bacteriophages with rapid release at the target site. Release of bacteriophage was studied under different preparation and pH conditions, demonstrating zero-order kinetics. It will be a solid contribution to the Polymers and will certainly appeal to many of its readers. I address some of the main issues with the manuscript in the next few paragraphs. The conclusions were verified by enough and convincing data. It is recommended that this manuscript to be published in Polymers after completing minor revision.
- PolyHIPE particles exhibited porosity, what are the unique advantages of this porous structure?
- About the hydrogel, more characterization should be done, such as the linear oscillatory frequency sweep experiment.
- Figure 4, the error bar should be added. Moreover, for equation fitting, R2 = 0.87, 0.89, the error is big. The authors should consider this point further.
- Introduction, the authors talked about drug delivery systems, the following recently published important related papers should be cited: Chem. Soc. Rev. 2017, 46, 7021; Chem. Soc. Rev. 2021, 50, 2839.
Reviewer 2 Report
Dear authors;
I think this contribution is an excellent article. I would recommend its acceptance after minor changes and few suggestions:
Line 94: there is no drug to be released. I think, the word “drug” must be removed.
In Section 2.5: Which temperature was set for the experiments? 37ºC? Please specify.
Line 168: …...the dispersion was mixed for 7 days...... The dispersion was mixed with what? I think, the right word should be “stirred”. Then:.....the dispersion was stirred for 7 days.... or .....the dispersion was kept under stirring for 7 days....
In Section 2.8. This equation was developed assuming the total amount of drug to be unchanged during the release process. Can we consider the total amount of bacteriophage to be constant? Please, explain.
Title of Section 3.2: Targeted drug release of bacteriophage....(there is no drug to be released). Can the bacteriophage be named as drug? If yes, please explain.
Figure 2: It is not clear. I would suggest to split the graphs in two figures. Each figure explaining a different experiment.
Figure 3: How was managed the error treatment?
Reviewer 3 Report
The manuscript polymers-1318150 investigate the polyHIPE/hydrogel system for targeted delivery of T7 bacteriophages with rapid release at the target site.
In my opinion, the manuscript has interesting results and I recommend the publication in Polymers journal, after major revisions.
- p.3, L. 144: Please explain the term “W/O/W”!
- p.5, L. 223: Please explain the term “HLB”!
- p.5, L. 205-217: Where did the authors used the Eq. (3) and (4) in the present manuscript? Did the authors establish “the dependence of the drug diffusion coefficient on the average mesh size”? Please make the correction!
- p.6, L. 231: Please add SEM images also for polyHIPE/hydrogel particles, with and without CaCl2 and discuss their porosity in comparison with polyHIPE and with the explanations from pag. 9!
- p.7, L. 266: The external graph from Figure 2 is a bit confusing! The time, t is expressed once in minutes and once in hours! Moreover, the Ct of the samples recorded an increase to 106 PFU/ml in the first 30 min, while at approximately 3.1 hours it drops again to about 105 PFU/ml! Please revise Figure 2 and make the adequate corrections!
- p.8, L. 311: Please explain the mechanism also for n=0.43, in function of the rates of penetrant diffusion and polymer chain relaxation!
- p.9, L. 350: In Figure 4 must be removed [s] from x-axis! In addition, please remove the equations from Figure 4 and add R2 in Table 1!
- p.10, L. 356: Why in Table 1 appear the biopolymer: 2% TOCNF and in Figure 4 is 1.5% TOCNF? Please make the adequate correction!
- p.9, L. 355: From the caption of Table 1 must be remove “obtained from Figure 4” and indicate the used sample!
- Please add a schematic representation of the polyHIPE/TOCNF synthesis, including the two cross-linking mechanisms of TOCNF hydrogels preparation, for a better understanding of the paper!
- Please add information related to the swelling degree of all samples!
- The Conclusions section presents only general data! Please improve this section!
